# SERS, XPS and DFT Study of Xanthine Adsorbed on Citrate-Stabilized Gold Nanoparticles

**DOI:** 10.3390/s19122700

**Published:** 2019-06-15

**Authors:** Stefano Caporali, Francesco Muniz-Miranda, Alfonso Pedone, Maurizio Muniz-Miranda

**Affiliations:** 1Department of Industrial Engineering, University of Florence, Via S. Marta 3, 50139 Firenze, Italy; stefano.caporali@unifi.it; 2Department of Chemical and Geological Sciences, University of Modena and Reggio Emilia, Via Campi 103, 41125 Modena, Italy; f.muniz-miranda@chimieparistech.psl.eu (F.M.-M.); alfonso.pedone@unimore.it (A.P.); 3Department of Chemistry “Ugo Schiff”, University of Florence, Via Lastruccia 3, 50019 Sesto Fiorentino, Italy

**Keywords:** SERS, XPS, DFT, gold nanoparticles, xanthine

## Abstract

We have studied the adsorption of xanthine, a nucleobase present in human tissue and fluids that is involved in important metabolic processes, on citrate-reduced gold colloidal nanoparticles by means of surface-enhanced Raman scattering (SERS), absorption, and X-ray photoelectron spectroscopy (XPS) measurements, along with density functional theory (DFT) calculations. The citrate anions stabilize the colloidal suspensions by strongly binding the gold nanoparticles. However, these anions do not impair the adsorption of xanthine on positively-charged active sites present on the metal surface. We have obtained the Fourier transform (FT)-SERS spectra of adsorbed xanthine by laser excitation in the near infrared spectral region, where interference due to fluorescence emission does not usually occur. In fact, the addition of chloride ions to the Au/xanthine colloid induces the aggregation of the gold nanoparticles, whose plasmonic band is shifted to the near infrared region where there is the exciting laser line of the FT–Raman instrument. Hence, this analytical approach is potentially suitable for spectroscopic determination of xanthine directly in body fluids, avoiding fluorescence phenomena induced by visible laser irradiation.

## 1. Introduction

In the last decade, gold nanoparticles have attracted much attention in the field of nanotechnology due to their many chemical and biomedical applications in sensing, catalysis, drug delivery, imaging, and nonlinear optical processes [1,2,3,4,5,6]. This is due to their peculiar properties, which can be summarized as: (1) marked chemical and physical stability; (2) high biocompatibility; (3) efficient surface functionalization with organic and biological ligands; and (4) enhanced optical responses related to surface plasmons. In particular, the use of gold nanoparticles in diagnostics, especially for the detection of DNA/RNA chains and their nucleobases [7,8,9], is getting more and more interest. The most common procedure for obtaining stable aqueous suspensions of gold nanoparticles is the reduction of tetrachloroauric acid by sodium citrate [10,11]. The role of the citrate anions in the formation of nanosized gold particles has been the subject of experimental and theoretical studies [12,13]. However, the type of interactions between citrate and gold nanoparticles, as well the effect on the stabilization and reactivity of the particles, have not been clarified yet.

Xanthine is a nucleobase present in human tissue and fluids that is involved in metabolic processes. Identifying the presence of xanthine in human organisms is a quite important task, because high levels of this molecule provoke severe problems by forming kidney stones as occurs with uric acid [14,15]. Surface-enhanced Raman scattering (SERS) spectroscopy is a powerful sensoristic technique as it overcomes the scarce sensitivity of the normal Raman technique, where molecules are adsorbed on silver or gold nanostructured substrates, with huge enhancement factors with respect to the Raman response of the non-adsorbed molecules [16,17]. Here, we have studied the adsorption of xanthine on citrate-reduced gold colloidal nanoparticles by means of SERS and X-ray photoelectron spectroscopy (XPS) measurements, along with density functional theory (DFT) calculations. In particular, we have analyzed the role of the citrate anions regarding both the metallic colloidal substrate and the xanthine molecules adsorbed on gold. In fact, a strong adsorption of citrate anions on gold could provide a double effect by (1) ensuring a good stabilization of the gold colloidal suspensions due to the repulsion between the negative charges, also in the presence of adsorbed organic molecules and (2) favoring the adsorption of ligand molecules by promoting the formation of suitable active-sites on the surface of the metal nanoparticles. Hence, this study is also of particular importance to understand the SERS response of these nanomaterials, which is closely related to the adsorption of ligand molecules. Moreover, with added chloride anions we have obtained the SERS-activation of the Au/xanthine colloids in the near-infrared spectral region, where fluorescence usually does not occur. Hence, this analytical approach is potentially suitable for SERS determination of xanthine directly in body fluids, avoiding fluorescence phenomena induced by visible laser irradiation.

## 2. Materials and Methods

### 2.1. Sample Preparation

By following the Turkevich preparation method [10] with tetrachloroauric acid (Aldrich, purity 99.9%) and sodium citrate (Aldrich, purity 99%), gold nanoparticles in colloidal aqueous suspension (pH ~6) were obtained. Xanthine (Sigma, purity ≥ 99.5%), purified by recrystallization, was added to the Au colloid (10^−6^ M or 10^−7^ M concentration). The addition of NaCl (10^−3^ M concentration) allowed observation of the Fourier transform (FT)-SERS spectra of xanthine.

### 2.2. Raman Measurements

FT-SERS spectra of Au/xanthine colloids in the Stokes spectral range of 50–3600 cm^−1^ were obtained with a Fourier transform (FT)-Raman spectrometer (Bruker Optics, Model MultiRam), a broad range quartz beam-splitter, an air-cooled Nd:YAG laser (1064 nm), and a Ge diode detector cooled with liquid nitrogen. The measurements were recorded in a 180° geometry, with 200 mW of laser power.

### 2.3. UV-Vis-NIR Absorption Measurements

Absorption spectra of the gold colloidal suspensions were observed at room temperature in the 200–1100 nm region with a Cary 5 Varian spectrophotometer by using 2 mm-optical pass quartz cuvettes.

### 2.4. XPS Measurements

Glass substrates with deposited Au colloidal nanoparticles were introduced in an ultrahigh vacuum (UHV; 2 × 10^−9^ mbar) system equipped with a VSW HAC 5000 hemispherical electron energy analyzer and a non-monochromatic Mg Kα X-ray source (1253.6 eV) through a loadlock under nitrogen flux, in order to minimize the exposure to air contaminants, and kept for at least 12 h before the measurements were taken. The XPS spectra, referenced to C 1s peak at 284.8 eV attributed to adventitious carbon, were acquired in the constant-pass-energy mode at Epas = 44 eV and fitted by means of CasaXPS software by adopting Gauss–Lorentz curves and subtraction of a Shirley-type background.

### 2.5. Computational Details

All DFT calculations were carried out by means of the Gaussian 09 package [18], adopting the B3LYP hybrid exchange and correlation functional [19,20] and LANL2DZ basis set [21,22,23]. The optimized geometries corresponded to true energy minima by considering that all vibrational frequencies were real and positive. The Raman intensities (*I_i_*) were obtained from the calculated activities (*A_i_*) by following the literature relationship [24,25]:
$$I_{i}=\frac{f{\left(\nu_{0}-\nu_{i}\right)}^{4}A_{i}}{\nu_{i}\left(1-e^{-\frac{hc\nu_{i}}{kT}}\right)}$$.

Here, *ν*_0_ and *ν_i_* are the excitation frequency and the *i*th vibrational frequency, respectively, expressed in cm^−1^, *h*, *c* and *k* are fundamental constants, and *f* is a common normalization factor for all peak intensities.

## 3. Results and Discussion

### 3.1. XPS Spectra

XPS spectra characteristics of carbon and gold, collected on soda-glass supported gold colloidal nanoparticles, are displayed in Figure 1 and Figure 2.

The spectral region of carbon 1s is complex and presents several constituents, at least 5, which can be reasonably attributed as follows:(1)C in aliphatic chains (deriving from atmospheric contamination and also from the CH_2_ group of citrates) at binding energy (BE) = 284.8 eV;(2)C-OH of citrate at BE = 286.4 eV, according to the literature data [26];(3)COO-Au at BE = 288.3 eV, whose binding energy is a little higher than that reported at 287.6 eV by Park et al. [27];(4)COOH (non-bonded to Au) at BE = 290.2 eV, analogously to what was reported by Park et al. [27];(5)C in carbonates at BE = 291.4 eV, according to the values (291.3–291.8 eV) reported in the literature [28]. The presence of carbonates could result from the dissolution of CO_2_ in the aqueous solvent during the reduction process of gold ions with citrate. It is reasonable that this signal was not observed in the work of Park [27], because in that case the gold nanoparticles were washed with a thiol solution with the removal of the most soluble components.

However, the XPS spectrum relating to gold is simpler (see Figure 2), so it can be interpreted by only two components, after removing the glass contribution at BE = 89.04 eV. The component at the lower BE (83.4 eV) is due to Au(0), whereas that at BE = 84.5 eV is due to Au(I). The peak relative to Au(0) occurs at a lower BE value than that usually found in nanosized metallic gold. For example, in gold nanoparticles obtained by laser ablation in pure water the peak due to Au(0) occurs at BE = 84.3 eV, as shown in the Appendix A. The shift observed in citrate-reduced gold nanoparticles cannot be related to instrumental calibration, because the BE values of carbon and glass (Si and O) well match those reported in the literature. In conclusion, the XPS spectra performed on gold nanoparticles show a large predominance of Au(0) strongly bound to citrate anions, which shift the binding energy of the metallic gold towards lower values, along with a smaller amount of Au(I).

Moreover, the presence of citrate anions linked to gold determines a negative zeta potential of the nanoparticles [29,30], preventing their aggregation and ensuring a marked stability in the aqueous suspension. In fact, even one year after the colloid preparation we did not observe any precipitate due to the collapse of metal nanoparticles.

### 3.2. UV-Visible-NIR Absorption Spectra

The adsorption of xanthine (10^−6^ M concentration) on citrate-reduced Au nanocolloid induces only a small shift of the plasmonic band maximum from 530 to 535 nm, without evidence of particle aggregation (see Figure 3). This means that the xanthine molecules do not replace the citrate ions, which remain bound to the metal surface, impairing particle aggregation due to the repulsion among the negative charges. Instead, for aggregation it is necessary to add NaCl (10^−3^ M concentration), which significantly increases the ionic strength of the aqueous suspension, promoting the formation of a secondary plasmonic band shifted to the near-infrared spectral region. Thus, it is possible to observe the SERS spectrum of xanthine by excitation in the near infrared (NIR) region, with a laser line at 1064 nm that is within the secondary plasmonic band, as shown in Figure 3.

### 3.3. FT-SERS Spectra

We have obtained FT-Raman spectra of xanthine adsorbed on Au colloidal nanoparticles by laser excitation in the near infrared (NIR) spectral region, where interference due to fluorescence does not generally occur. By excitation at 1064 nm, no distinct SERS signal of xanthine (10^−6^ M concentration) is detected, unless NaCl is added. Instead, in the presence of chloride anions (10^−3^ M concentration), a strong FT-SERS spectrum is observed thanks to the plasmonic band shift towards the NIR region. This SERS spectrum, shown in Figure 4, is dominated by the most intense band at 658 cm^−1^, along with other intense bands at higher frequencies. The adopted concentration of xanthine in the gold colloids is quite low (10^−6^ M) in order to test the SERS efficiency with biological samples, where the presence of xanthine is relatively scarce. However, we have also obtained a SERS spectrum with 10^−7^ M xanthine, as reported in the Appendix A. This spectrum appears quite similar to that reported in Figure 4 with 10^−6^ M xanthine, albeit weaker with increased noise. In the Appendix A the SERS spectrum (10^−6^ M) is compared with the Raman spectrum of xanthine in water solution at the same pH (around 6). Xanthine is barely soluble in water, so its concentration in solution is 10^−3^ M. By multiplying the Raman spectrum of xanthine by ten, comparable intensities are shown for both SERS and Raman bands. Hence, the adsorption of xanthine on citrate-reduced gold nanoparticles allows at least a Raman enhancement of four orders of magnitude to be obtained.

### 3.4. DFT Calculations

For a complete interpretation of the SERS data, a DFT computational approach is necessary, because in solution xanthine is presumably present as a mixture of two tautomers, N7(H) and N9(H) [31,32], whose structures are reported in the inset of Figure 4. DFT calculations, concerning the structures of the metal/molecule complexes that are formed on nanostructured metal surfaces and their Raman response, allow the acquisition of a large amount of information on the adsorption of organic ligands, along with the provision of a correct vibrational assignment of the observed SERS bands [33,34,35]. In the case of xanthine adsorbed on gold nanoparticles, we could identify which tautomer binds to metal and by which molecular site, and what interaction is established between the molecule and the active sites of the metal surface, along with the characteristics of the latter. Therefore, we have taken into account the binding of both tautomers with gold through the lone-pairs of the sp^2^ nitrogen atoms of the imidazolic ring of xanthine, i.e., the N9 atom of the tautomer N7(H) and the N7 atom of the tautomer N9(H). These atoms can act as electron-donors in the formation of metal/molecule complexes. For the interaction with gold, Au° or Au^+^ were considered as metal active sites (adatoms), on the basis of the indications deriving from XPS spectra, which show the surface presence of Au(0) with a minority amount of Au(I). Observed and calculated frequencies are compared in Table 1, whereas the optimized structures of the four complexes taken into account are shown in Figure 5. From the DFT calculations, only the N9(H)-Au^+^ complex is able to reproduce all the frequencies observed in the SERS spectrum. Instead, the complexes with Au(0) are completely unsuitable for predicting the SERS bands. In this regard, it should be noted that the interactions of xanthine with Au^+^ provoke a much larger electronic charge-transfer from molecule to metal than the interactions with Au°, with shorter N-Au bonds (see Figure 5). In particular, the N9(H)-Au^+^ complex shows the strongest electronic charge-transfer, as reported in Table 1. Mulliken and Hirshfeld partial charges in the N7(H)-Au^+^ and N9(H)-Au^+^ complexes are reported in Appendix A. Finally, while the tautomer N7(H) is more stable than the tautomer N9(H), the latter bound to Au^+^ results more stable than the corresponding complex of the tautomer N7(H) with Au^+^. This is due to the fact that in the N9(H)-Au^+^ complex, in addition to the N7-Au bond, an interaction is also established with the vicinal oxygen atom, as shown in Figure 5.

A further indication concerning the adsorption of xanthine on gold can be deduced from the simulated SERS spectra of the complexes, as reported in Appendix A: The N9(H)-Au^+^ complex shows a SERS profile that better reproduces the experimental one with respect to that corresponding to the N7(H)-Au^+^ complex.

Finally, on the basis of our DFT calculations, the correct assignment of the observed SERS bands is now possible, as shown in Figure 6. All the prominent SERS bands are in-plane deformations. In particular, the dominant SERS band at 658 cm^−1^ is attributable to the breathing vibration of the pyrimidinic ring of the molecule, whereas those at higher wavenumbers to deformations that involve ring bond stretching, H-bending, and C=O stretching modes. This assignment is important for identifying the “marker” SERS bands of xanthine in order to detect this nucleobase when other organic components are present in the sample. In particular, this analytical approach can be adopted to recognize xanthine in the presence of other purine nucleobases (adenine, guanine), which show a different spectral pattern. The SERS spectrum of adenine adsorbed on gold is dominated by a very strong band at 735 cm^−1^ [36,37], very far from the strongest SERS band of xanthine at 658 cm^−1^, without any spectral coincidence with the marker bands of xanthine at 961 and 1709 cm^−1^. It is also possible to distinguish the SERS spectrum of xanthine from that of guanine adsorbed on gold [38], even if SERS bands at 665, 958 and 1705 cm^−1^ are present in the latter, because the marker bands of xanthine at 1248 and 1320 cm^−1^ do not find correspondence in the SERS spectrum of guanine. A comparison between the marker SERS bands of xanthine, guanine, and adenine is shown in the Appendix A.

## 4. Conclusions

The present study indicates that xanthine adsorbs on gold nanoparticles as tautomer N(9)H, linked to surface adatoms modelled as Au^+^ in the DFT simulations. This conclusion is similar to that obtained from the adsorption of xanthine on Ag colloidal nanoparticles [39]. The citrate anions are strongly bound to metallic gold, as evidenced by the XPS measurements, but they allow the adsorption of ligand molecules. Indeed, these citrate anions impair the aggregation of the gold nanoparticles, also in the presence of adsorbed xanthine molecules, due to repulsion among the negative charges, and promote the formation of positively-charged active sites on the metal surface, which are able to bind the ligand molecules. The addition of chloride ions to the Au/xanthine aqueous suspension promotes the aggregation of the gold nanoparticles, ensuring the observation of the FT-SERS spectrum by excitation in the NIR region, where fluorescence usually does not occur. Hence, this analytical approach is potentially suitable for SERS determination of xanthine directly in body fluids, avoiding spectral interference induced by visible laser irradiation. In actuality, fluorescence emission can be due to trace impurities that impair the observation of the Raman bands, as often occurs in biological samples [40]. Moreover, as the NIR region is coincident with the biological tissue transparency window [41], this could allow the use of SERS spectroscopy to recognize the presence of xanthine both in vitro and in vivo. 

Finally, the identification of the marker SERS bands of xanthine could allow the spectroscopic recognition of this molecule in biological samples, where other components such as purine nucleobases are present.

## Figures and Tables

**Figure 1 sensors-19-02700-f001:**
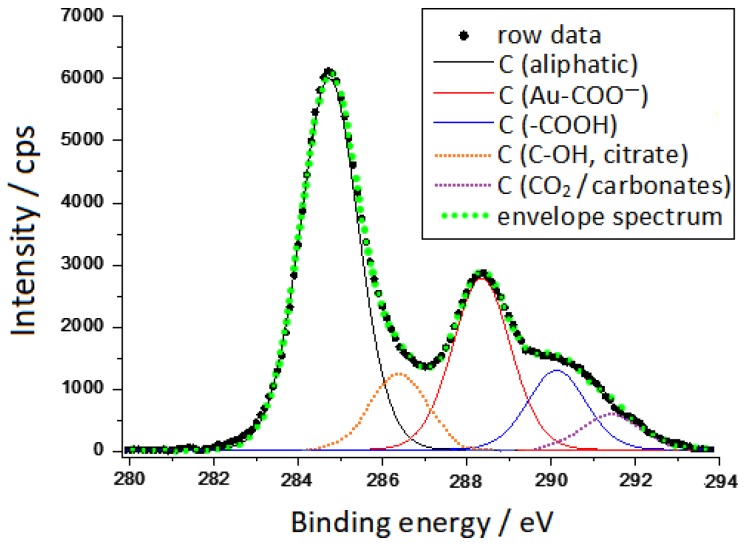
X-ray photoelectron spectroscopy (XPS) spectrum of deposited Au nanoparticles in the carbon 1s spectral region.

**Figure 2 sensors-19-02700-f002:**
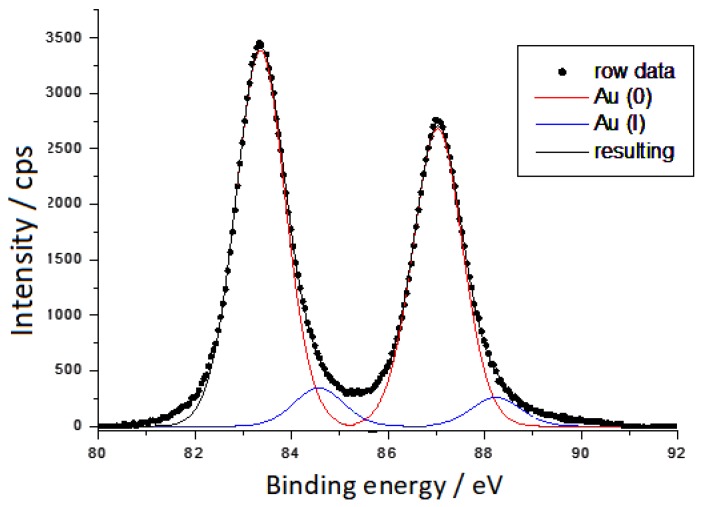
Au 4f doublet XPS spectral region of deposited Au nanoparticles.

**Figure 3 sensors-19-02700-f003:**
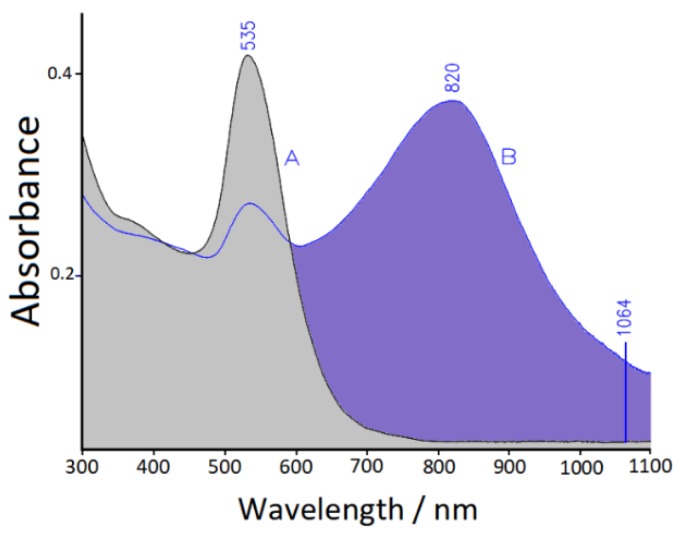
UV-vis-near infrared (NIR) extinction spectrum of Au colloids with 10^−6^ M xanthine, before (A) and after (B) the addition of 10^−3^ M NaCl. The 1064 nm excitation laser line is indicated.

**Figure 4 sensors-19-02700-f004:**
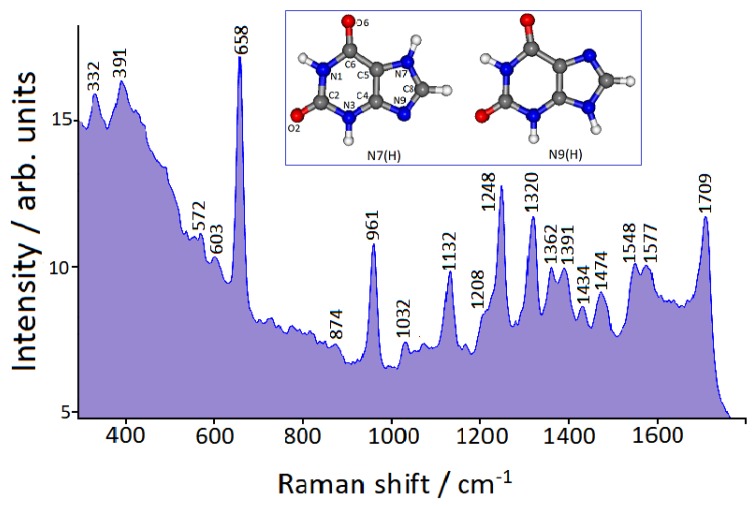
Fourier transform-surface-enhanced Raman scattering (FT-SERS) spectrum of 10^−6^ M xanthine in Au colloid (1064 nm exciting line). The tautomeric forms of xanthine are shown in the inset.

**Figure 5 sensors-19-02700-f005:**
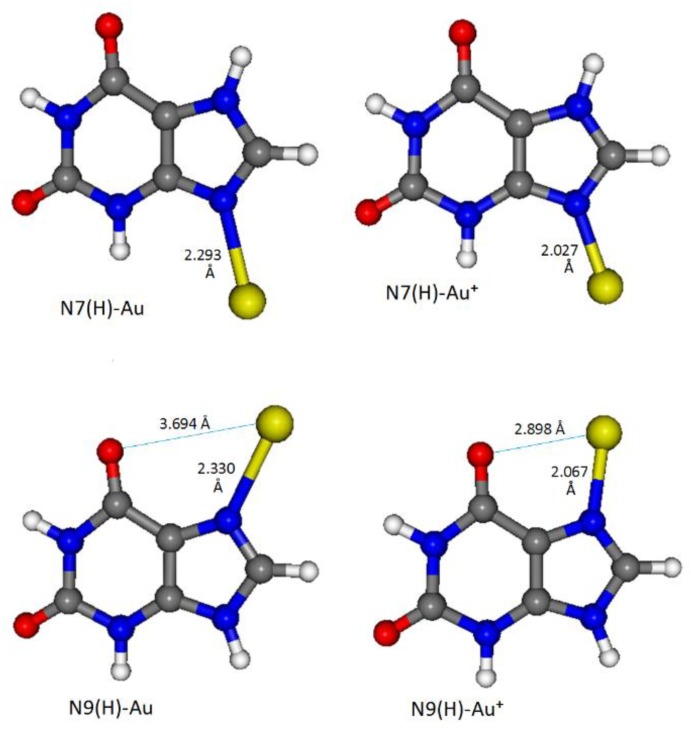
Density functional theory (DFT)-optimized structures of the xanthine/gold complexes.

**Figure 6 sensors-19-02700-f006:**
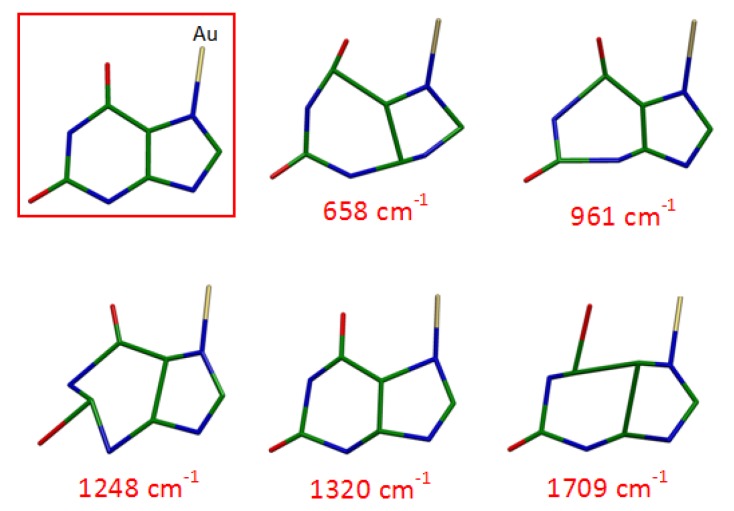
Normal modes of the N9(H)-Au^+^ complex, corresponding to the prominent SERS bands of xanthine adsorbed on gold.

**Table 1 sensors-19-02700-t001:** Observed and calculated surface-enhanced Raman scattering (SERS) frequencies (cm^−1^).

SERS on Au Nanoparticles	Calc. N7(H)-Au°	Calc. N9(H)-Au°	Calc. N7(H)-Au^+^	Calc. N9(H)-Au^+^
332	342	332	344	352
391	371	365	381	375
572		548	572	589
603	613	611	619	617
658	677	675	659	653
874	873	841	855	857
961	971	963	954	950
1032			1013	1014
1132	1121	1121	1130	1132
1208	1207		1218	1230
1248	1270	1281	1269	1252
1320	1310	1310	1334	1326
1362	1322	1379		1346
1391	1408	1421	1415	1415
1434	1422	1435	1433	1428
1474	1464		1464	1458
1548			1502	1515
1577	1584	1573	1591	1588
1709	1712	1720	1720	1702
energy (a.u.)	−697.8012	−697.7842	−697.5511	−697.5584
charge-transfer ^1^	−0.1910 *e*	−0.1763 *e*	−0.3279 *e*	−0.3544 *e*
charge-transfer ^2^	−0.1938 *e*	−0.1916 *e*	−0.5308 *e*	−0.5361 *e*

^1^ based on Mulliken partial charges; ^2^ based on Hirshfeld partial charges; *e* = proton charge.

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
