# Peer review of "SERS, XPS and DFT Study of Xanthine Adsorbed on Citrate-Stabilized Gold Nanoparticles"

_sensors, 2019, doi:10.3390/s19122700_

Round 1
Reviewer 1 Report
This paper deals on comprehensive analysis about the xanthine adsorbed on citrate-stabilized gold nanoparticle by SERS, XPS, and DFT study. The topic of this study may be of interests to many readers. In addition, the fundamental findings about interaction between xanthine and citrate-stabilized gold nanoparticles may be applicable to the detection of xanthine directly in body fluids by NIR irradiation. Therefore, I suggest accept this manuscript without further consideration.
Author Response
I am pleased that Reviewer 1 appreciated our paper.
Reviewer 2 Report
According to the revision of the manuscript, I summarize the following points to reconsider:
The abstract seems confusing, for example "The SERS-activation of the gold 25 nanoparticles in the near-infrared spectral region, where a spectral 26 interference due to fluorescence usually does not occur, has been obtained by 27 addition of chloride anions." , is this description correct? Or restructure.
Regarding format, the figures are not unified. For example, figure 2 and figure 3 differ in the title of their X and Y axes. You should read the instructions to adapt all the figures to the format of the journal.
Why was not a comparison made with Raman spectroscopy between Xanthine without gold nanoparticles, and Xanthine in gold nanoparticles ?. This in order to verify the levels of intensity (factor of amplification) between one and another sample. Besides that in figure 4 I consider it important to add the numerical values for the Y axis.
According to the title of the manuscript "SERS, XPS and DFT study of Xanthine adsorbed on citrate-stabilized gold nanoparticles", I believe that more evaluations should have been done regarding molar concentrations of Xanthine deposited in gold nanoparticles.
In the same way, it will be interesting to verify both simulated (as shown in the supplement material) and experimentally changes in Raman intensity when different concentrations are used (gold or xanthine).
Verify the format of the references, in specific reference 4 and 33.
Review the manuscript in the grammar part of the English text.
Author Response
1) In the abstract we have clarified the relationship between the chloride anions added to the Au/xanthine colloid and the SERS-activation in the near infrared region.
2) We have unified the formats of Figure 1, Figure 2 and Figure 3. In Figure 4 we have added the numerical values for the Y axis.
3) Now we have reported in the Supplementary Materials (Figure S3) a comparison between the SERS spectrum of xanthine (10-6M) and the Raman spectrum of xanthine in water solution at the same pH (around 6). Xanthine is scarcely soluble in water, so the concentration in solution is 10-3 M. By multiplying the Raman spectrum of xanthine by ten, comparable intensities are shown for both SERS and Raman bands. Hence, the adsorption of xanthine on citrate-reduced gold nanoparticles allows obtaining at least a Raman enhancement of four orders of magnitude. This point is now reported in the Results.
4) The adopted gold concentration is that reported in the literature (ref. 10) for the preparation of citrate-reduced gold nanoparticles, because it provides both colloidal stability and adsorption capability. The adopted concentration of xanthine in the gold colloid is quite low (10-6M), in order to verify the SERS efficiency in biological samples, where the presence of xanthine is relatively scarce. However, we have obtained also a SERS spectrum with 10-7M xanthine, as reported now in the Supplementary Materials (Figure S2), albeit weaker with increased noise. This point is now reported in the Results.
5) We have verified the format of the References.
6) The grammar part of the English text has been reviewed.
Reviewer 3 Report
Miranda et. al describe the detection of xanthine absorbed on gold nanoparticles using SERS and XPS. I find that the work lacks novelty, and does not include any significant progress in the design of new sensors for the detection metabolites. The authors do not characterize xanthine beyond merely using SERS or XPS. There is no information about the concentration or dose-dependent detection. I do not see how relevant UV-Vis spectra is to determine the absorption of xanthine. The authors need to significantly improve the design of the sensor and a rationale for using citrate based gold nanoparticles for the detection of xanthine. They should also consider the detection xanthine using SERS in the presence of other nucleotides.
Author Response
The aim of this paper is not centered on the design of new sensors, but on these points:
1) The XPS characterization of citrate-reduced gold nanoparticles, which was not previously performed in this detail.
2) The elucidation of the role played by the citrate anions in the stabilization of the gold nanoparticles, as well as in the ligand adsorption.
3) The information by SERS and DFT studies on the tautomer of xanthine adsorbed on gold, as well as on the adsorption molecular sites and the nature of the metal active-sites.
4) The use of the SERS spectroscopy by excitation with 1064-nm laser line, by shifting the plasmonic band of the gold nanoparticles towards the near-infrared region as a result of an aggregation process induced by added chloride anions. For this point, the use of the UV-vis-NIR absorption spectroscopy is essential.
Using FT-SERS spectroscopy by excitation in the NIR region could allow the detection of xanthine directly in body fluids, avoiding the spectral interference of fluorescence, which often impairs the observation of the Raman bands of biological samples when the excitation occurs in the visible region. In this respect, the identification of the “marker” bands of xanthine in the SERS spectrum (see Figure 6) is a quite important point, because it allows the spectroscopic recognition of this molecule also in the presence of other purine nucleobases (adenine, guanine), which show a different spectral pattern. The SERS spectrum of adenine adsorbed on gold is dominated by the very strong band at 735 cm-1 (refs. 36,37), very far from the strongest SERS band of xanthine at 658 cm-1, without any spectral coincidence with the marker bands of xanthine at 961 and 1709 cm-1. It is also possible to distinguish the SERS spectrum of xanthine from that of guanine adsorbed on gold (ref. 38), even if SERS bands at 665, 958 and 1705 cm-1 are there present, because the marker bands of xanthine at 1248 and 1320 cm-1 do not find correspondence in the SERS spectrum of guanine. Now a comparison between the marker SERS bands of xanthine, guanine and adenine adsorbed on gold is shown in the Supplementary Materials (Table S2). These considerations are now reported in the Conclusions.
The adopted concentration of xanthine in the gold colloids is quite low (10-6M), in order to verify the SERS efficiency in biological samples, where the presence of xanthine is relatively scarce. However, we have obtained also a SERS spectrum with 10-7M xanthine, as reported now in the Supplementary Materials (Figure S2). This spectrum appears quite similar to that reported in Figure 4 with 10-6M xanthine, albeit weaker with increased noise. This point is now reported in the Results.
Now we have also reported in the Supplementary Materials (Figure S3) a comparison between the SERS spectrum of xanthine (10-6M) and the Raman spectrum of xanthine in water solution at the same pH (around 6), similar to that physiological. Xanthine is scarcely soluble in water, so the concentration in solution is 10-3M. By multiplying the Raman spectrum of xanthine by ten, comparable intensities are shown for both SERS and Raman bands. Hence, the adsorption of xanthine on citrate-reduced gold nanoparticles allows obtaining at least a Raman enhancement of four orders of magnitude. This point is now reported in the Results.
The grammar part of the English text has been reviewed
Round 2
Reviewer 2 Report
Thank you for attending most of the observations that were made to you. Emphasize the format to present the figures.
Best Regards
Author Response
I am pleased that the Reviewer has appreciated our paper. We have emphasized in the text the presence of the Figures. Best regards.
Reviewer 3 Report
The authors need to consider improving the introduction section, especially the rationale behind studying the interactions between xanthine and citrate coated nanoparticles, and how an understanding of these interactions will have a relevance to SERS Spectra of Xanthine.
Author Response
Following the Reviewer’s suggestion, in the Introduction as well as in the Conclusion we have better explained (highlighting in red) the relationship between citrate anions and xanthine adsorption, in order to emphasize its importance in the SERS response.